

# Diagnostic accuracy of the atherogenic index of plasma in metabolic syndrome: a diagnostic meta-analysis

Yuge Gao[1], Chengcheng He[1] and Jia Mi[2]

[1] Changchun University of Chinese Medicine, Changchun, Jilin, China
[2] Affiliated Hospital, Changchun University of Chinese Medicine, Changchun, Jilin, China

## ABSTRACT

**Objective**. This study aims to assess the diagnostic performance of the atherogenic index of plasma (AIP) in estimating the risk of metabolic syndrome (MetS) among adults across various geographic regions.

**Methods**. A comprehensive search was conducted across EMBASE, Web of Science, PubMed, and the Cochrane Library from their inception until September 29, 2024. Eligible studies were selected and evaluated for methodological quality using the Quality Assessment of Diagnostic Accuracy Studies-2 (QUADAS-2) framework. Statistical analyses were performed using STATA 15.1. Sensitivity, specificity, diagnostic odds ratio, positive and negative likelihood ratios, the summary receiver operating characteristic (SROC), curve, and 95% confidence intervals (CI) were calculated to assess the diagnostic accuracy of AIP for MetS. Statistical significance was defined as a $p$-value < 0.05.

**Results**. Eleven observational studies involving 36,463 participants were included. The analysis showed that AIP is an effective biomarker for identifying the risk of MetS, with an area under the curve (AUC) of 0.84 (95% CI [0.81–0.87]). AIP demonstrated comparable diagnostic value in both males and females, with an AUC of 0.82, highlighting its potential utility in sex-specific assessments. Geographic region, diagnostic reference standards, and publication year were identified as potential sources of heterogeneity.

**Conclusions**. This study demonstrates that AIP is a relatively accurate tool for detecting MetS, supporting its role in prevention and in reducing the risk of associated chronic diseases. Further research with larger sample sizes and multi-center designs is needed to explore the combined use of AIP with other biomarkers to enhance diagnostic accuracy for MetS.

# INTRODUCTION

Metabolic syndrome (MetS) is a group of contributing factors affecting metabolic processes, collectively increase the likelihood of cardiovascular disease, type 2 diabetes, and various types of cancer (*Ribeiro et al., 2018*). The worldwide prevalence of MetS among adults varies between 12.5% and 31.4%, depending on the diagnostic criteria applied, with the

Corresponding author
Jia Mi, mijia0101@126.com

most notable rates observed in the Eastern Mediterranean region and the Americas. Its prevalence tends to elevate as national income levels rise (*Dobrowolski et al., 2022*). As of 2020, around 3% of children and 5% of adolescents were impacted by MetS (*Noubiap et al., 2022*). In China, a multi-center cross-sectional study reported a MetS detection rate of 19.58%, with 26.97% in men and 16.18% in women (*Li et al., 2016*). Definitions of MetS vary slightly across countries and organizations, including the World Health Organization (WHO) criteria, the National Cholesterol Education Program Adult Treatment Panel III (NCEP-ATP III) guidelines, and the International Diabetes Federation (IDF) standards (*Jiang et al., 2024*). In 1998, the WHO introduced a standardized definition to unify the concept of MetS and provide a practical diagnostic tool for clinicians and researchers (*Hsu, Kuo & Lai, 2024*). These criteria include impaired fasting glucose, diabetes, abnormal glucose tolerance, or insulin resistance, along with at least two additional parameters, such as abnormal body measurements, elevated blood pressure, or lipid profiles. However, the complexity of these criteria—requiring multiple disease diagnoses and various physical and biochemical measurements—makes the diagnostic process for MetS cumbersome and costly, hindering its rapid clinical identification. Consequently, exploring quick and reliable diagnostic markers for MetS remains a pressing challenge.

Metabolic syndrome and oxidative stress exhibit a tightly intertwined bidirectional pathological link: Visceral adipose tissue accumulation and insulin resistance significantly elevate reactive oxygen species (ROS) levels through the excessive release of free fatty acids (*Pliouta et al., 2025*), activation of NADPH oxidases, and mitochondrial dysfunction. Concurrently, hyperglycemia-induced glycative stress further amplifies oxidative damage (*Pingitore et al., 2024*). These processes collectively trigger lipid peroxidation, inflammatory cascades, and suppression of antioxidant defenses (*e.g.*, downregulation of the Nrf2 pathway), accelerating the progression of complications such as cardiovascular disease, type 2 diabetes, and cancer—ultimately forming a self-perpetuating vicious cycle (*Jia, Hill & Sowers, 2023*). Oxidative stress functions both as a core pathological byproduct of metabolic syndrome and a critical mediator in driving its clinical complications (*Padgett et al., 2023*).

The atherogenic index of plasma (AIP) is an emerging composite lipid marker employed to evaluate the risk of atherosclerosis and heart disease (*Okan et al., 2024*). It is computed by taking the logarithm of the proportion of triglycerides (TG) to high-density lipoprotein cholesterol (HDL-C) in the bloodstream (*Fernández-Aparicio et al., 2022*; *Qu et al., 2024*). AIP not only indicates the equilibrium between beneficial and atherogenic lipoproteins but also functions as a powerful indicator of atherosclerosis and coronary artery disease (*Demirtola et al., 2024*). Recently, researchers have scrutinized the diagnostic performance of AIP for MetS. However, studies evaluating its diagnostic performance for MetS have yielded inconsistent results. For instance, one study reported that AIP exhibited excellent diagnostic performance for MetS (AUC: 0.914) (*Vega-Cárdenas et al., 2023*), while another study found a lower diagnostic performance (AUC: 0.716) (*Zhang et al., 2021*). These discrepancies underscore the need for further investigation into the applicability and accuracy of AIP across diverse populations (*Lee et al., 2016*).

Given the existing controversies in published studies, this study aims to comprehensively investigate and validate the clinical value and diagnostic capability of AIP for MetS through a comprehensive review and meta-analysis. By providing an in-depth scientific analysis of the diagnostic performance of AIP, this study seeks to enhance early detection mechanisms for MetS and provide evidence-based insights for healthcare professionals and public health policymakers. The findings are expected to significantly improve the accuracy of early disease detection and establish a solid scientific foundation for subsequent clinical interventions and chronic disease risk management.

## MATERIALS AND METHODS

This research was documented in the International Prospective Register of Systematic Reviews (PROSPERO) under the identifier CRD42024603143. The study adhered to the Preferred Reporting Items for Systematic Reviews and Meta-Analyses of Diagnostic Test Accuracy Studies (PRISMA-DTA) guidelines (*Salameh et al., 2020*).

### Literature search strategy

A comprehensive search was carried out across electronic databases, including EMBASE, Web of Science, PubMed and Cochrane, from their establishment until September 29, 2024, to identify studies eligible for this research. Literature screening was completed on October 31, 2024.The search utilized keywords such as ''atherogenic index of plasma'', ''atherogenic index'', ''metabolic syndrome'', and ''syndrome of insulin resistance''. Additionally, reference lists of pertinent articles and gray literature were manually searched to identify further eligible research. Detailed search strategies are provided in Table S1.

#### *Inclusion and exclusion criteria*

Studies had to fulfill the following PECOS criteria. (1) **P**articipants: General population suspected of having metabolic syndrome, with no restrictions on gender or age. (2) **E**xposure: Studies that included AIP measurements. (3) **O**utcomes: Studies reporting diagnostic outcomes, including true positives (TP), true negatives (TN), false positives (FP), and false negatives (FN). (4) **C**omparators: Studies with a clearly defined diagnostic gold standard. (5) **S**tudy design: Case-control studies, observational cohort studies (prospective or retrospective), and cross-sectional studies. Only peer-reviewed journal articles published in English-language were considered.

The exclusion criteria were as follows: (1) Non-English language studies. (2) Studies that did not calculate AIP cutoff values associated with metabolic syndrome risk factors. (3) Studies based on duplicate data from the same survey or investigation. (4) Protocols, editorials, abstracts, or conference proceedings.

#### *Data extraction*

Two researchers (Gao and Mi) independently screened the studies. The data extraction process was completed on December 3, 2024. Titles and abstracts were reviewed to exclude irrelevant studies. Potentially eligible studies were assessed through full-text review to confirm their eligibility. Any disputes were rectified through discussion or by involving a third researcher (He) to reach a consensus.

Data extraction was independently carried out by these researchers, utilizing a pre-designed electronic form. Extracted data included the first author, study design, country or region of the study, year of publication, diagnostic gold standard used, sample size, gender distribution (male/female), age, thresholds, detection techniques, sensitivity and specificity. This rigorous process ensured data accuracy and study reliability.

*Quality assessment*

The quality of the methodology in the included studies was assessed by two researchers (Gao and Mi) independently implementing Review Manager 5.4 and (QUADAS-2) tool (*Whiting et al., 2011*). The QUADAS-2 framework encompasses four key areas: reference standard, index test, patient selection, and flow and timing. Each of these areas contained specific questions designed to help assessors judge the risk of bias.

*Statistical analysis*

The diagnostic accuracy of AIP in diagnosing MetS was assessed by calculating sensitivity, specificity, diagnostic odds ratio, positive and negative likelihood ratios, SROC curve, and 95% confidence intervals (CI). Heterogeneity was measured with Cochrane's $I^2$ statistics, where $P < 0.10$ or $I^2 > 50\%$ indicated significant heterogeneity. In such cases, a random-effects model was leveraged, and subgroup analyses were implemented. To evaluate publication bias, funnel plots were utilized, and Deeks' funnel plot asymmetry test was applied quantitatively, with substantial significance indicated by $P < 0.05$. All statistical analyses were executed using STATA 15.1 software, and results were considered statistically relevant when the *P*-value was less than 0.05.

# RESULTS

Through electronic searches, we originally collected 2,250 relevant articles. Once duplicates were removed and the abstracts and titles were reviewed, we selected 20 studies for full-text eligibility review. During this process, nine articles were excluded for not meeting the inclusion criteria. Ultimately, the systematic review and meta-analysis included a total of 11 studies. The PRISMA flowchart clearly illustrates the various stages and the number of studies selected, as depicted in Fig. 1.

This study included 11 cross-sectional studies, covering 36,463 adolescents and adults aged 13 years and older. This studies were conducted in China, India, Iran, Mexico, and Ghana—representing populations from Asia, Africa, and Europe. The sample size varied between 250 and 9,904 participants. The characteristics of the studies, optimal cutoff values, and diagnostic performance are summarized in Table 1. The critical AIP values ranged from 0.02 to 1.0, distributed as follows: two studies used cutoff values between 0.02 and 0.07, two between 0.449 and 0.49, another two studies had values between 0.449 and 0.49, and seven between 0.50 and 1.0. Additionally, three studies reported cutoff values greater than 1.

## Quality assessment

Bias risks and applicability concerns were assessed in terms of patient selection, index test, reference standard, and timing. The majority of studies had a bias risk rated as "low" or

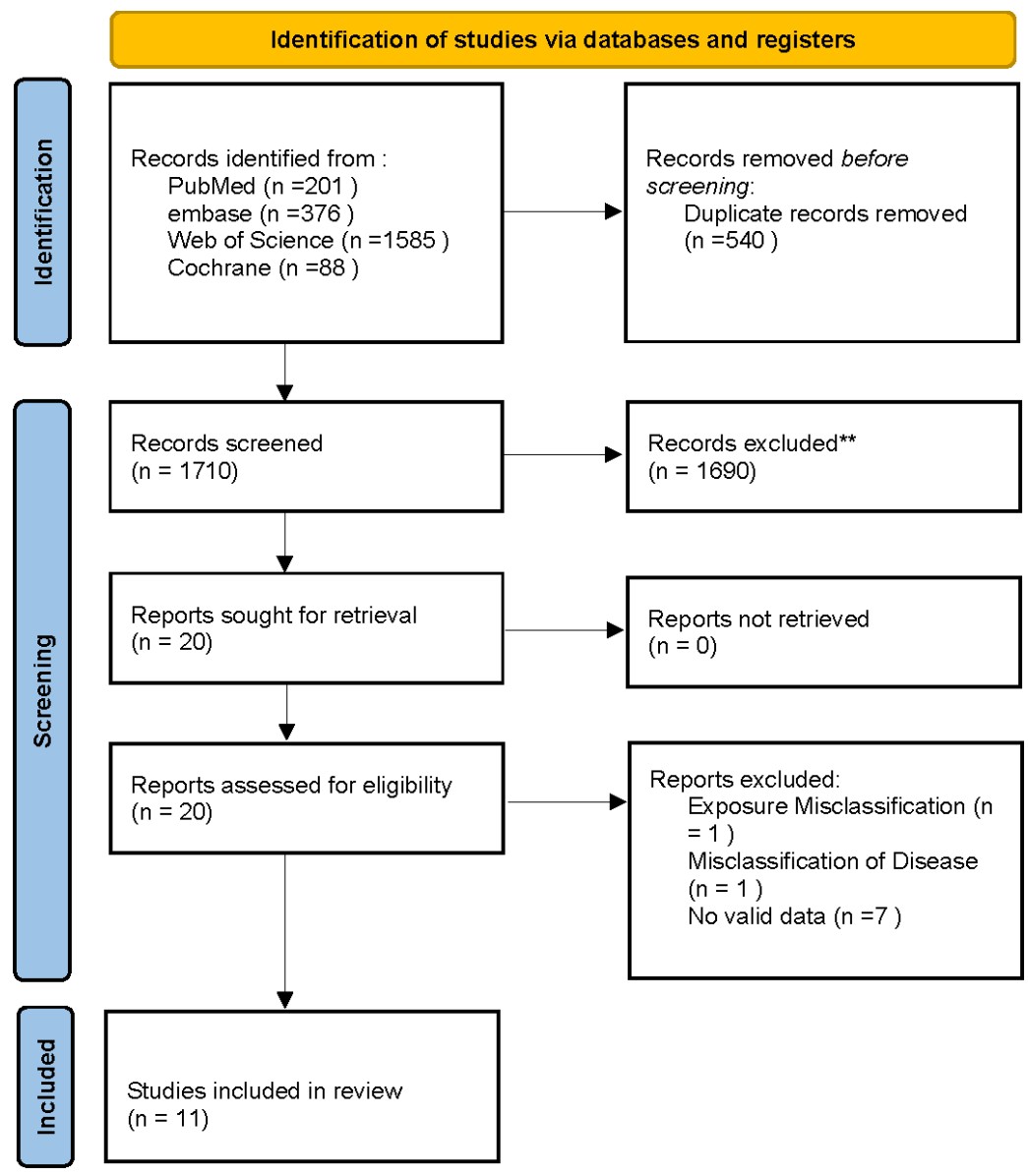

**Figure 1** **PRISMA flowchart for systematic review and meta-analysis: literature search and study selection.**

"unclear", suggesting a certain risk of bias in the design and execution. Several studies exhibited high quality with a low risk of bias, as illustrated in Figs. 2A and 2B.

## Combined results and additional analyses

In the overall analysis, the combined sensitivity was 0.77 (95% CI [0.72–0.81]), and the specificity was 0.84 (95% CI [0.81–0.87]). The positive likelihood ratio (LR+) was 3.41 (95% CI [2.9–4.33]), and the negative likelihood ratio (LR-) was 0.27 (95% CI [0.2–0.37]). The SROC curve was 0.79 (95% CI [0.73–0.84]), as presented in Figs. 3A, 3B and 4.

**Table 1** Main characters of the studies included in the meta-analysis.

| Authors | Year | Area | Study design | Participants | Reference standard | TP | FP | FN | TN |
|---|---|---|---|---|---|---|---|---|---|
| Huiying Liang | 2013 | China | cross-sectional study | 4,706 | JIS criteria | 63 | 120 | 10 | 788 |
| Yu-Tung Tien | 2023 | Taiwan | cross-sectional study | 128 | Alberti et al. | 44 | 18 | 7 | 59 |
| M Sabarinathan | 2022 | India | case-control study | 300 | IDF criteria | 135 | 14 | 15 | 136 |
| Xiaocui Chen | 2016 | China | cross-sectional study | 494 | IDF criteria | 843 | 956 | 377 | 1,918 |
| Bang-Dang Chen | 2015 | China | cross-sectional study | 4,767 | IDF criteria | 1,211 | 822 | 366 | 2,368 |
| Hossein Babaahmadi-Rezae | 2024 | Iran | cross-sectional study | 9,809 | IDF criteria | 3,369 | 1,430 | 1,116 | 3,894 |
| Xianghui Zhang | 2021 | China | cross-sectional study | 9,904 | IDF criteria | 1,744 | 1,763 | 802 | 4,595 |
| Angel Fernandez-Aparicio, RN | 2022 | Spanish | cross-sectional study | 981 | IDF criteria | 63 | 120 | 10 | 788 |
| Xiang-Hui Zhang | 2016 | China | cross-sectional study | 3,752 | IDF criteria | 529 | 944 | 274 | 2,005 |
| Mariela Vega-Cardenas | 2023 | Mexico | cross-sectional study | 1,372 | AHA criteria | 260 | 243 | 24 | 845 |
| Fareed K N Arthur | 2012 | Ghana | cross-sectional study | 250 | IDF criteria | 83 | 27 | 7 | 143 |

**Notes.**

*Fernández-Aparicio et al. (2022), Chen et al. (2015), Arthur et al. (2012), Babaahmadi-Rezaei et al. (2024), Liang et al. (2013), Vega-Cárdenas et al. (2023), Sabarinathan et al. (2022), Zhang et al. (2016), Zhang et al. (2021), Chen et al. (2016), Tien et al. (2023).*

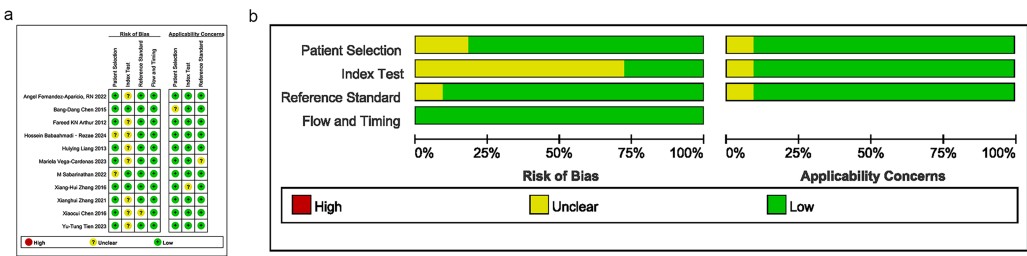

**Figure 2** Results of literature quality assessment; (A) domain and overall risk of bias; (B) weighted bar chart for risk of bias assessment. Note: *Fernández-Aparicio et al. (2022), Chen et al. (2015), Arthur et al. (2012), Babaahmadi-Rezaei et al. (2024), Liang et al. (2013), Vega-Cárdenas et al. (2023), Sabarinathan et al. (2022), Zhang et al. (2016), Zhang et al. (2021), Chen et al. (2016), Tien et al. (2023).*

Considerable heterogeneity was detected across the studies ($I^2 > 90\%$). The diagnostic odds ratio (dOR) was 12.74 (95% CI [7.41–21.91]), as presented in Table 2.

For studies involving female patients, the integrated sensitivity, specificity, and AUC were 0.73 (95% CI [0.67–0.79]), 0.77 (95% CI [0.72–0.82]), and 0.82 (95% CI [0.78–0.85]), respectively. The LR+ was 3.23 (95% CI [2.39–4.38]), while the LR- was 0.33 (95% CI [0.23–0.46]). Significant heterogeneity in sensitivity and specificity was found ($I^2 > 90\%$), and the Q statistic was significant ($df = 4.00$, $p = 0.00$). The (dOR) was 5 (95% CI [5–13]), as presented in Table 2.

For male patients, the integrated sensitivity, specificity, and AUC were 0.73 (95% CI [0.67–0.79]), 0.77 (95% CI [0.72–0.82]), and 0.82 (95% CI 0.78 to 0.85), respectively. The LR+ was 2.73 (95% CI [1.79–4.17]), and the LR- was 0.33 (95% CI [0.26–0.46]). High heterogeneity was observed in sensitivity and specificity ($I^2 > 90\%$), with a significant Q statistic ($df = 4.00$, $p = 0.00$). The dOR was 5 (95% CI [5–13]), as presented in Table 2.

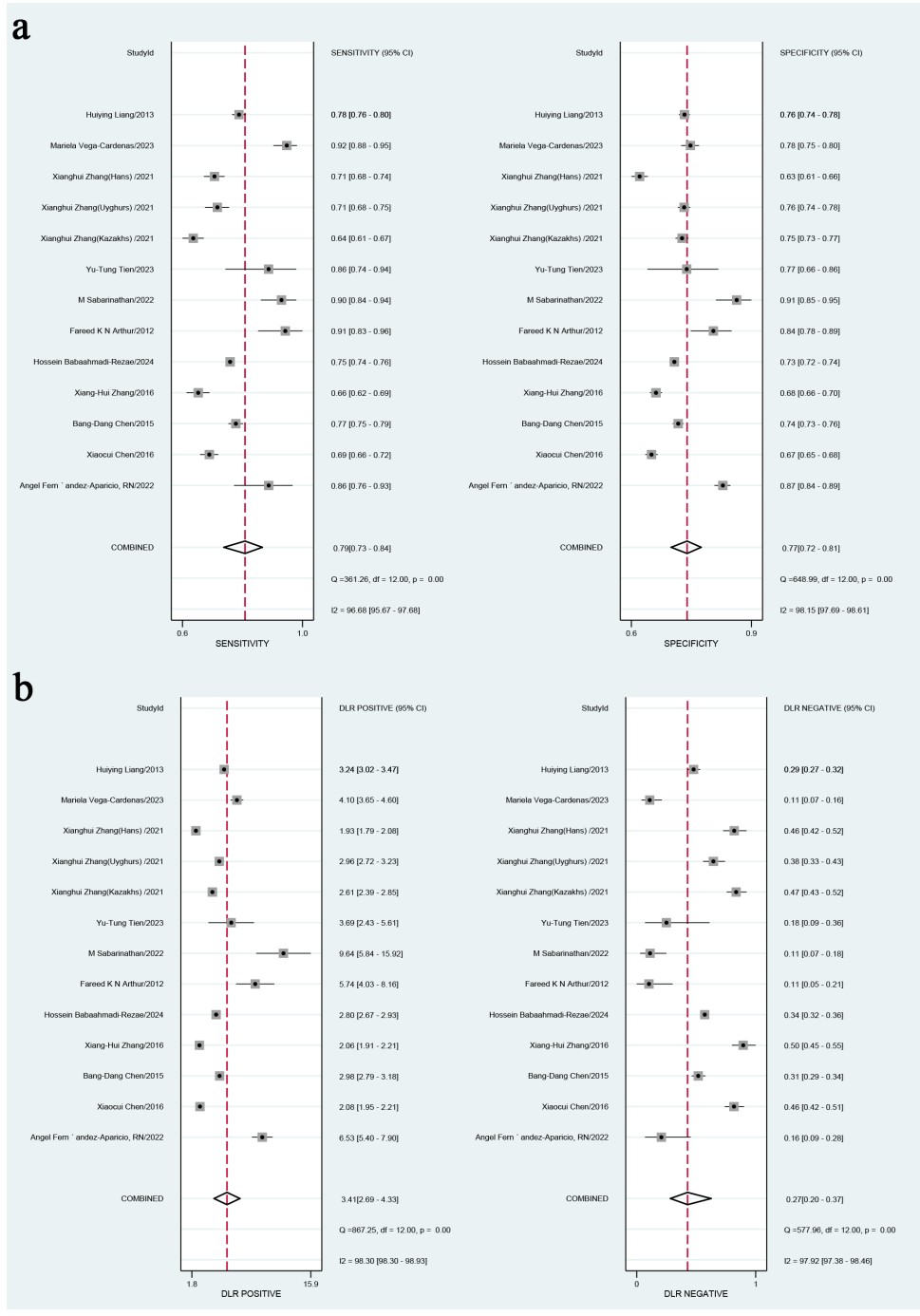

**Figure 3** **(A) Combined sensitivity and specificity in the overall study; (B) positive likelihood ratio and negative likelihood ratio.** Note: *Fernández-Aparicio et al. (2022)*, *Chen et al. (2015)*, *Arthur et al. (2012)*, *Babaahmadi-Rezaei et al. (2024)*, *Liang et al. (2013)*, *Vega-Cárdenas et al. (2023)*, *Sabarinathan et al. (2022)*, *Zhang et al. (2016)*, *Zhang et al. (2021)*, *Chen et al. (2016)*, *Tien et al. (2023)*.

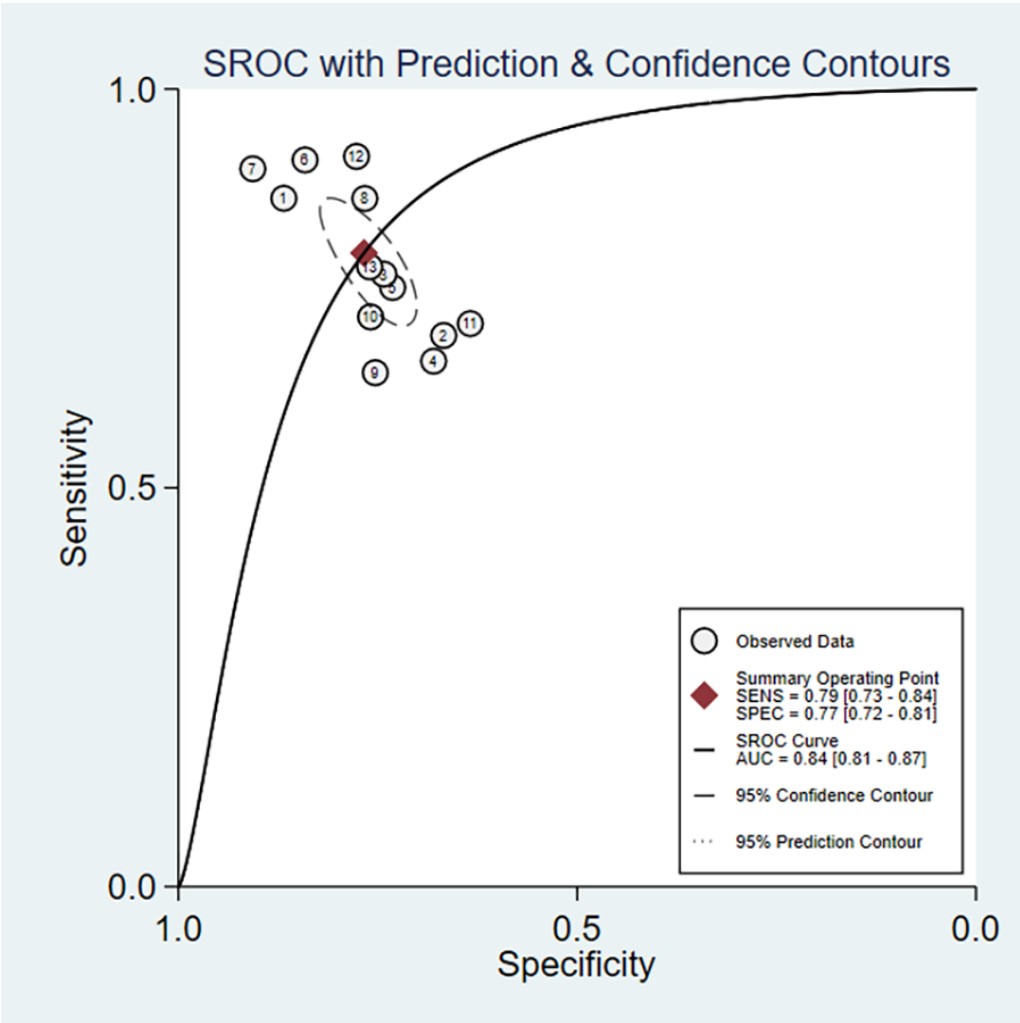

**Figure 4   Area under the curve (AUC) of the receiver operating characteristic (ROC) curve for atherogenic index of plasma in diagnosing metabolic syndrome.**

## Subgroup analysis

In the subgroup analysis, several factors were identified as potential factors affecting the variability of sensitivity and specificity. For sensitivity, both the reference standard and geographical region showed statistical significance ($p < 0.001$), suggesting that these two factors may contribute to the heterogeneity of combined sensitivity. Additionally, publication year also showed significance ($p < 0.01$), indicating that the year of publication may affect heterogeneity. Patient age and sample size did not show statistical significance, suggesting that these factors were not potential sources of heterogeneity in sensitivity.

For specificity, geographical region, patient age, and publication year were recognized as significant contributors to heterogeneity ($p < 0.001$). Reference standard and sample size also showed statistical significance ($p < 0.01$), implying that these factors also influenced specificity, though to a lesser extent than geographical region, patient age, and publication

**Table 2 Sensitivity, specificity, area under the receiver operating characteristic curve, positive likelihood ratio, negative likelihood ratio, diagnostic odds ratio, and publication bias for atherogenic index of plasma in diagnosing metabolic syndrome.**

| Population | Studies (n) | Sensitivity | Specificity | AUSROC | LR+ | LR- | dOR | Publication bias (*p*-value) |
|---|---|---|---|---|---|---|---|---|
| Overall | 11 | 0.79 [0.73, 0.84] | 0.77 [0.72, 0.81] | 0.84 [0.81, 0.87] | 3.4 [2.7, 4.31] | 0.27 [0.20, 0.37] | 13 [7, 22] | 0.097 |
| Male | 5 | 0.76 [0.71, 0.81] | 0.72 [0.61, 0.81] | 0.81 [0.77, 0.84] | 2.7 [1.8, 4.2] | 0.33 [0.23, 0.46] | 8 [4, 18] | 0.55 |
| Female | 5 | 0.73 [0.67, 0.79] | 0.77 [0.72, 0.81] | 0.82 [0.78, 0.85] | 3.2 [2.4, 4.4] | 0.34 [0.26, 0.46] | 9 [5, 17] | 0.998 |

| Subgroup | Category | Chi$^2$ | P (sen)-value | P (spe)-value |
|---|---|---|---|---|
| AREA | Outside of Asia | Reference | | |
| | Asia | 12.19 | <0.001 | <0.001 |
| AGE | <18 | | | |
| | ≥18 | 3.73 | 0.16 | <0.001 |
| Gold standard | NOT IDF criteria | | | |
| | IDF criteria | 4.01 | <0.001 | <0.001 |
| Publication year | after 2020 | | | |
| | Before 2020 | 0.84 | <0.001 | |
| Sample size | >500 | | | |
| | <500 | 4.88 | 0.12 | <0.001 |

year. Consequently, it was imperative to consider these potential sources of heterogeneity when interpreting the results, as detailed in Table S1.

## Publication bias assessment

The points corresponding to studies were symmetrically distributed in the Deeks' funnel plots for overall, male, and female populations. The results of Deeks' test revealed no significant publication bias ($P = 0.1$, $P = 0.55$, $P = 0.99$), as depicted in Figs. 5A–5C.

## DISCUSSION

In recent years, the prevalence of MetS has been steadily increasing, highlighting the urgent need for effective and rapid identification methods (*Bhurosy & Jeewon, 2014*; *Caballero, 2007*; *Glas et al., 2003*). Based on our knowledge, this is the first meta-analysis to quantify the diagnostic performance of the AIP for MetS across diverse populations and to evaluate differences between sexes.

A growing body of research has validated plasma AIP as a biomarker reflecting the degree of atherogenicity. AIP has been identified as a significant predictor of diabetes, MetS, non-alcoholic fatty liver disease (NAFLD) and atherosclerosis (*Cure et al., 2018*; *Wang et al., 2018*). AIP also serves as an alternative indicator for low-density lipoprotein (LDL) small and dense particles. Elevated AIP levels indicate a higher probability of oxidized particles and foam cell formation, which can lead to increased oxidized apolipoprotein B and LDL-C levels (*Li et al., 2021*). Persistently high AIP levels are generally correlated with sustained high triglyceride (TG) levels and/or relatively low high-density lipoprotein cholesterol (HDL-C) levels (*Qin & Chen, 2024*). As TG levels rise, they compete with glucose for cellular uptake, the quantity and function of insulin receptors on adipocytes,

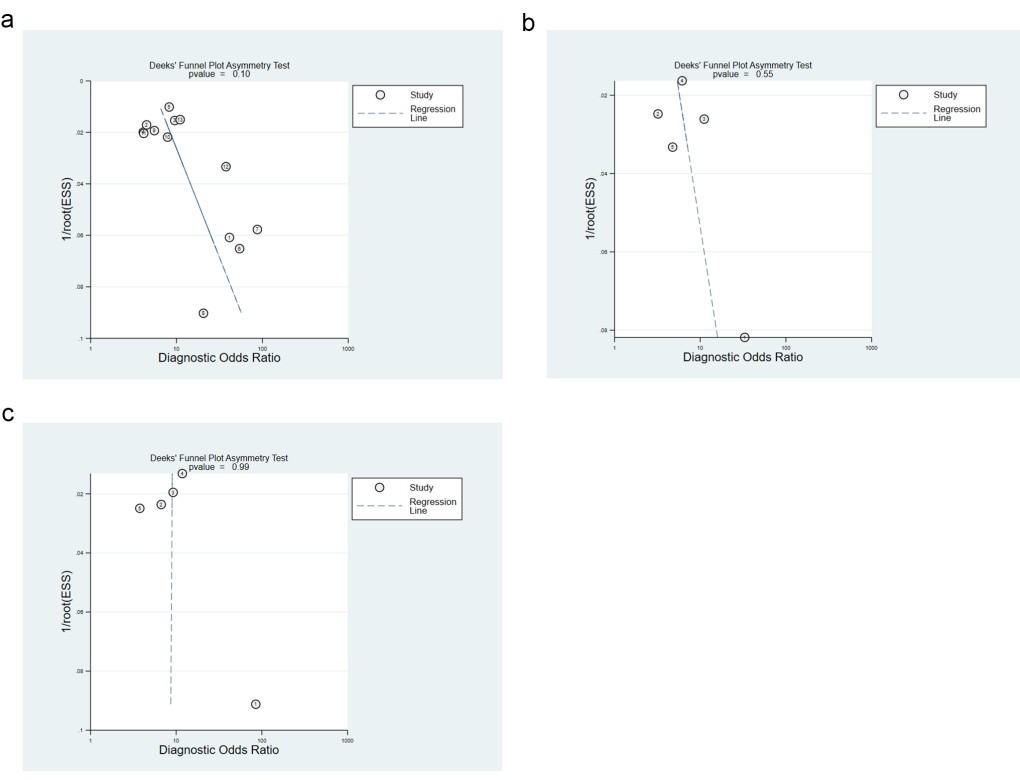

**Figure 5** (A) Overall publication bias assessment using deek's funnel plot; (B) publication bias assessment for males using deek's funnel plot; (C) publication bias assessment for females using deek's funnel plot.

interfering with normal insulin binding. Notably, AIP has a high sensitivity in anticipating acute coronary events, and its prognostic capacity for cardiovascular diseases (CVD) has been firmly established (*Dobiásová, Urbanová & Samánek, 2005*). One of the advantages of AIP is its wide measurable range, spanning from negative to positive values, with zero nearly linked to the 25.5 nm diameter of LDL-C, a critical threshold identified in earlier studies as the boundary between LDL-C phenotypes A and B (*Njajou et al., 2009*). Furthermore, elevated TG levels prompt the increased free fatty acids and the generation of harmful lipids, disrupting insulin pathway and causing overabundant glucagon release. Concurrently, low HDL-C levels impair cholesterol reverse transport, leading to cholesterol accumulation in pancreatic β-cells, resulting in β-cell dysfunction, decreased insulin secretion, elevated blood glucose levels, and β-cell apoptosis (*Daniels et al., 2008*).

Clinically, our robust evidence (AUC 0.84) supports AIP as a cost-effective triage tool. Although gender differences may influence AIP's cutoff values and diagnostic performance, potentially due to hormonal levels, fat distribution, and other biological factors (*Dobiásová & Frohlich, 2001*; *Superko, 1996*). There are intrinsic differences in blood lipid metabolism between men and women. In premenopausal women, estrogen helps regulate blood lipids, leading to relatively higher HDL-C levels and lower TG levels, which result in lower AIP values. In contrast, in men, androgen predominates, leading to relatively lower HDL-C

levels and higher TG levels, resulting in higher AIP values (*Jung & Choi, 2014*). Men tend to have higher waist circumference, smoking rates, and alcohol consumption compared to women. However, according to the results of this study, the diagnostic percision of AIP does not differ significantly between genders, further supporting its diagnostic value for MetS across sexes.

The significant clinical value of the AIP in detecting MetS necessitates gender-differentiated treatment strategies (*Zou et al., 2023*). For example, an AIP > 0.21 indicates high risk in females, compared to >0.24 in males. Postmenopausal estrogen decline significantly increases AIP sensitivity in women. Consequently, therapeutic priorities for females include enhancing HDL functionality (*e.g.*, Omega-3 supplementation) and addressing insulin resistance (*Moussavi Javardi et al., 2020*). Conversely, due to the higher prevalence of hypertriglyceridemia in males, treatment should focus on intensifying glycemic control, reducing TG levels (*e.g.*, with fenofibrate), and promoting weight loss through (requiring strict limitation of alcohol and refined carbohydrate intake). Core lifestyle interventions encompass resistance training to augment muscular glucose uptake, combined with a Mediterranean diet supplemented with flaxseed to improve HDL function. Additionally, high-intensity interval training (HIIT) is particularly recommended for its efficacy in rapidly reducing visceral adipose tissue, alongside a low-carbohydrate diet to effectively mitigate postprandial TG surges (*Shahiddoust & Monazzami, 2025*).

The findings of this study contribute to improving diagnostic tools for MetS, particularly for rapid identification (*Chen et al., 2024*). By providing evidence-based guidance, the study also can help clinicians and public health experts diagnose and manage MetS more effectively. Additionally, the study suggests that AIP may be equally effective for gender-specific diagnoses of MetS, providing a foundation for future gender-tailored diagnostic strategies (*Graham, 2023*).

Nevertheless, this study has several limitations. First, AIP cutoff values for identifying MetS are not universally applicable. Variations in participant age, gender, and regional backgrounds across different studies may cause bias, compromising the accuracy of the findings (*N, Shankar & Narasimhappa, 2020*).

Despite these limitations, the study presents several key strengths that enhance its scientific and clinical impact. First, to our knowledge, this is the first meta-analysis to comprehensively evaluate the diagnostic accuracy of AIP for MetS across diverse global populations, addressing a critical gap in cardiovascular risk stratification. Second, our analysis incorporated robust methodological rigor through adherence to PRISMA-DTA guidelines, application of QUADAS-2 quality assessment, and utilization of a large pooled sample ($n = 36,463$) spanning multiple continents, enhancing it's statistical power and generalizability. Third, the demonstration of comparable diagnostic efficacy in both genders (AUC 0.82 for males and females) despite established biological differences in lipid metabolism, underscores AIP's unique value as a sex-neutral triage tool. Finally, the identification of geographic region and diagnostic criteria as sources of heterogeneity provides actionable insights for developing population-specific AIP thresholds.

While the study covers various countries from Asia, Africa, and Europe, differences in sample sizes may limit the generalizability of the findings. Moreover, data from certain

regions or populations may be underrepresented. The study acknowledges that different gender and age groups may require distinct AIP cutoff values, but insufficient data were provided to determine these differences (*Mao et al., 2024*). Significant heterogeneity was observed in the combined sensitivity and specificity, which could diminish the validity of the meta-analysis findings (*Haidar et al., 2024*). Future studies should explore AIP cutoff values for specific gender and age groups, which are crucial for personalized diagnosis and treatment of MetS. Furthermore, multicenter intervention research is necessary to examine the effect of lowering AIP levels in preventing MetS, providing scientific evidence for the development of new prevention strategies.

## CONCLUSION

This meta-analysis evaluated the diagnostic accuracy of AIP in identifying the risk of MetS among adults from diverse regions worldwide. The results demonstrate that AIP is a reliable diagnostic indicator for MetS. Future research should focus on customizing AIP cutoff values, expanding sample sizes, tracking AIP changes over time, exploring the pathological mechanisms linking AIP and MetS, and assessing the impact of lowering AIP levels in preventing MetS. Additionally, integrating other biomarkers to improve diagnostic accuracy will be crucial. These efforts will enhance our understanding of MetS and help optimize diagnosis and treatment strategies.

### Funding
This research was not supported by any specific grant from funding agencies in the public, commercial, or not-for-profit sectors.

### Competing Interests
The authors declare there are no competing interests.

### Author Contributions
- Yuge Gao conceived and designed the experiments, performed the experiments, analyzed the data, prepared figures and/or tables, authored or reviewed drafts of the article, and approved the final draft.
- Chengcheng He conceived and designed the experiments, performed the experiments, analyzed the data, prepared figures and/or tables, and approved the final draft.
- Jia Mi conceived and designed the experiments, performed the experiments, analyzed the data, prepared figures and/or tables, and approved the final draft.

### Data Availability
This is a systematic review/meta-analysis.
## Supplemental Information

Supplemental information for this article can be found online at http://dx.doi.org/10.7717/peerj.20074#supplemental-information.

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
