# Peer review of "Diagnostic accuracy of the atherogenic index of plasma in metabolic syndrome: a diagnostic meta-analysis"

_PeerJ, doi:10.7717/peerj.20074_

## Round 0.1 · original submission · Major Revisions

·

Basic reporting

Diagnostic accuracy of the atherogenic index of plasma in patients with metabolic syndrome: a diagnostic meta-analysis
Manuscript
The article ‘Diagnostic accuracy of the atherogenic index of plasma in patients with metabolic syndrome: a diagnostic meta-analysis’
This research aims to assess and determine the diagnostic performance of the atherogenic index of plasma (AIP) in estimating the risk of metabolic syndrome (MetS) among adults across various geographic regions.
This is an interesting and remarkable study that aims to assess and determine the diagnostic performance of the atherogenic index of plasma (AIP) in estimating the risk of metabolic syndrome (MetS) among adults across various geographic regions. I believe it will be a better text after the corrections I suggested.
Comments to the author(s)
1- This study demonstrated that the AIP is relatively accurate in the detection of MetS, promoting the prevention of MetS and reducing the risk of associated chronic diseases. With this inference, should there be a difference in the treatment approach to these patients? My humble suggestion would be to briefly state your opinions on this matter.
2- What do you think your study brings new to the literature?
3- Spelling errors in the article should be corrected and spelling rules should be observed.

Experimental design

This research aims to assess and determine the diagnostic performance of the atherogenic index of plasma (AIP) in estimating the risk of metabolic syndrome (MetS) among adults across various geographic regions.
This is an interesting and remarkable study that aims to assess and determine the diagnostic performance of the atherogenic index of plasma (AIP) in estimating the risk of metabolic syndrome (MetS) among adults across various geographic regions. I believe it will be a better text after the corrections I suggested.

Validity of the findings

This is an interesting and remarkable study that aims to assess and determine the diagnostic performance of the atherogenic index of plasma (AIP) in estimating the risk of metabolic syndrome (MetS) among adults across various geographic regions. I believe it will be a better text after the corrections I suggested.
This study demonstrated that the AIP is relatively accurate in the detection of MetS, promoting the prevention of MetS and reducing the risk of associated chronic diseases.

Reviewer 2 ·

Basic reporting

I received for review an original research article entitled "Diagnostic Accuracy of the Atherogenic Index of Plasma in Patients with Metabolic Syndrome: A Diagnostic Meta-Analysis", prepared by Yuge Gao, which was submitted to the PeerJ. Cardiometabolic disorders are one of the most important challenges for public health in the modern world. Metabolic syndrome is a set of disorders contributing to increased cardiovascular risk and the development of overt cardiovascular disease. Cardiovascular diseases, especially in the course of atherosclerosis, are the main cause of morbidity and mortality in many countries of the world. Developing diagnostic methods that will better identify patients at increased risk of cardiometabolic diseases is therefore extremely important. In my opinion, the manuscript presents a certain scientific value. However, some significant changes are necessary, which may contribute to increasing the value and attractiveness of the presented manuscript.

1) The introduction is written quite well. The authors draw attention to how big a problem metabolic syndrome is, as well as to the difficulties associated with the existing controversies on this topic. At the beginning of the introduction, the authors also draw attention to the most important complications of metabolic syndrome, such as heart-related diseases, type 2 diabetes, and various cancers. I propose to mention here that obesity and metabolic syndrome are also associated with increased oxidative stress, which may play a significant role in the pathogenesis of the diseases mentioned. I also propose to replace the term "heart-related diseases" with the term "cardiovascular diseases".

2) The following formulation appears in the discussion: "a biomarker reflecting the degree of plasma atherosclerosis". This should be corrected. There is no such thing as "plasma atherosclerosis". I assume it should be "atherogenicity".

3) In another place in the discussion, the phrase "AIP has the maximum sensitivity in anticipating acute coronary events" appears. In my opinion, this formulation is not correct. It would be more appropriate to say "AIP has a high sensitivity in anticipating acute coronary events".

4) Not only limitations but also strengths of the study should be discussed.

Experimental design

No additional comments.

Validity of the findings

No additional comments.

Additional comments

No additional comments.

---

## Round 0.2 · accepted · Accept

Dear authors,

After a thorough review of the manuscript, the reviewers have no further comments. Therefore, your manuscript is ready for publication.

Reviewer 2 ·

Basic reporting

I received for review a revised version of the original research article entitled "Diagnostic Accuracy of the Atherogenic Index of Plasma in Patients with Metabolic Syndrome: A Diagnostic Meta-Analysis", prepared by Yuge Gao, which was submitted to PeerJ. The paper has been improved. I have no further critical comments.

Experimental design

-

Validity of the findings

-